



# The Berkeley Environmental Air-quality and $CO_2$ Network: field calibrations of sensor temperature dependence and assessment of network scale $CO_2$ accuracy

Erin R. Delaria[1], Jinsol Kim[2], Helen L. Fitzmaurice[2], Catherine Newman[1], Paul J. Wooldridge[1], Kevin Worthington[1], and Ronald C. Cohen[1,2]

[1]Department of Chemistry, University of California Berkeley, Berkeley, CA 94720, USA
[2]Department of Earth and Planetary Science, University of California Berkeley, Berkeley, CA 94720, USA

**Correspondence:** Ronald C. Cohen (rccohen@berkeley.edu)

**Abstract.**

The majority of global anthropogenic $CO_2$ emissions originate in cities. We have proposed that dense networks are a strategy for tracking changes to the processes contributing to urban $CO_2$ emissions and suggested that a network with ~2 km measurement spacing and ~1 ppm node-to-node precision would be effective at constraining point, line and area sources within cities.

Here we report on an assessment of the accuracy of the Berkeley Environmental Air-quality and $CO_2$ Network (BEACO$_2$N) $CO_2$ measurements over several years of deployment. We describe a new procedure for improving network accuracy that accounts for and corrects the temperature dependent zero offset of the Vaisala CarboCap GMP343 $CO_2$ sensors used. With this correction we show that a total error of 1.6 ppm or less can be achieved for networks that have a calibrated reference location and 3.6 ppm for networks without a calibrated reference.

## 1   Introduction

The atmosphere has warmed approximately $1 \pm 0.2$ °C since pre-industrial times, which is unequivocally due to anthropogenic emissions of $CO_2$ and other greenhouse gases (GHGs) (IPCC, 2013). Global initiatives are needed to limit warming to 1.5 °C by achieving net zero GHG emissions by 2050 and a 45% emissions decline from 2010 levels by 2030 (Rogelj et al., 2018). As over 70% of global anthropogenic $CO_2$ emissions originate from cities (United Nations, 2011) , effective $CO_2$ monitoring

strategies in urban regions are needed to assess progress toward emissions commitments.

Monitoring trends in $CO_2$ emissions by tracking ambient $CO_2$ in urban environments is challenging because of the large diversity of emissions sources, complex spatial and temporal patterns of emission rates, varied topography and the effects of meteorology on the observed concentrations (e.g. Vardoulakis et al., 2003; Lateb et al., 2016). As a result, most cities rely exclusively on economics and social data and do not check that their reported emissions match the observed $CO_2$ enhancements

in the air over their city. To date, most efforts to assess $CO_2$ emissions from cities have relied upon a small number of high-cost $CO_2$ instruments that provide precise and accurate representations of regional signals. Other approaches include use of correlations between $CO_2$ and other gases, measurements of [14]C in annual grasses, and use of satellite column $CO_2$ observations



such as from OCO-2 (e.g. Pataki et al., 2006; Riley et al., 2008; Thompson et al., 2009; Kort et al., 2013; Andrews et al., 2014; Fu et al., 2019; Ye et al., 2020). Most of these efforts have used as a target metric an annual average of fossil fuel-related $CO_2$

emissions from an entire city (e.g. McKain et al., 2012; Kort et al., 2013; Bréon et al., 2015; Verhulst et al., 2017). Simultaneous measurements of CO and $^{14}CO_2$ have also provided information about sector-specific emission sources (Turnbull et al., 2015). Other methods of evaluating urban emissions have relied on emissions inventories (e.g. Gurney et al., 2009; Gately et al., 2013, 2017). These emissions inventories are frequently applied to inverse modelling approaches in combination with either short-term mobile measurements or a small number of long-term measurement sites to extract regional emissions (e.g. Brondfield

et al., 2012; Sargent et al., 2018; Nathan et al., 2018; Turnbull et al., 2019). Several studies have also combined a network of $CO_2$ observations with inverse modelling approaches to evaluate the accuracy of emissions inventories and $CO_2$ sources (e.g. Lauvaux et al., 2016, 2020).

We are pursuing a distinct approach aimed at process level understanding of the components of an urban emissions inventory. To do so, we are developing tools for deployment of spatially dense networks of $CO_2$ measurements, in combination with

gases and aerosols that are co-emitted and that affect air quality. The result is an ability to map emissions with ∼1 km or "neighborhood scale" fidelity. The Berkeley Environmental Air-quality and $CO_2$ Network (BEACO$_2$N) (Turner et al., 2020; Kim et al., 2018; Shusterman et al., 2018, 2016; Turner et al., 2016) is our platform for research and development of tools for dense networks. New deployments in Glasgow, Scotland, and Los Angeles, California are bringing new collaborators and experience in different cities to the project. BEACO$_2$N has been operating since 2012 in the San Francisco Bay area and

consists of over 70 nodes separated by approximately 2 km (Fig. 1). The nodes incorporate commercially available, low-cost sensors for measuring CO, NO, $NO_2$, $O_3$, particulates, and $CO_2$.

Turner et al. (2016) assessed the performance of a hypothetical BEACO$_2$N-like observing system coupled to an inverse model and demonstrated that a random measurement uncertainty of 1 ppm between nodes was adequate to meaningfully constrain $CO_2$ emissions from a point, line, or area source of 147, 45, and 9 tC hr$^{-1}$, respectively. With 1 ppm network uncertainty,

weekly $CO_2$ emissions in the San Francisco Bay area could be estimated to within 5% error. In this paper we describe advances in our approach to maintaining stable, multi-year comparability among BEACO$_2$N nodes in a city and evaluate the accuracy achieved with these new procedures. Our emphasis in the revised approach to sensor accuracy is on tracking and correcting the temperature dependence of the Vaisala CarboCap GMP343 $CO_2$ instruments. We present development and evaluation of the methods using observations from the San Francisco Bay Area BEACO$_2$N deployment and then apply these ideas to the

BEACO$_2$N network in Houston, Texas.

## 2   Development of a $CO_2$ field calibration method for Vaisala temperature dependence

The efficacy of a network of a large number of low cost non-dispersive infrared (NDIR) $CO_2$ sensors to evaluate $CO_2$ emissions has been previously discussed (Shusterman et al., 2016; Turner et al., 2016; Martin et al., 2017; Shusterman et al., 2018; Müller et al., 2020). Martin et al. (2017) showed that after correcting six SenseAir K30 carbon dioxide NDIR sensors (with off-the-

shelf reported errors of 5-20 ppm) for environmental variables, the median root mean square error could be reduced to below





2 ppm, making the sensors potentially useful for ambient air-quality monitoring. Recently, Müller et al. (2020) evaluated the potential applications of a low-cost $CO_2$ NDIR sensor network for resolving site-specific $CO_2$ signals in Switzerland. The calibration method of Müller et al. (2020) involved laboratory chamber calibrations of over 300 low-cost NDIR $CO_2$ sensors and ambient co-location with a reference instrument prior to deployment, as well as regular monitoring and drift correction

during a 2-year deployment period. Shusterman et al. (2016) developed an *in situ* method for calibrating and correcting for individual instrument biases and temporal drifts of the Vaisala CarboCap GMP343 $CO_2$ instruments deployed in the BEACO$_2$N nodes. Using this method, Shusterman et al. (2018) demonstrated that the BEACO$_2$N network could provide highly sensitive detection of changes to traffic emissions at a scale relevant to policy concerns. Shusterman et al. (2018) also illustrated the efficacy of the BEACO$_2$N network in showing both regional $CO_2$ emissions and local $CO_2$ enhancements at the scale of a

single neighborhood. In an analysis of the BEACO$_2$N observations for 6 weeks before and after the COVID-19 shutdown, Turner et al. (2020) showed that a 25% change in emissions is easily derived by an inverse model and that hourly variations in emissions can be inferred.

The use of a large number of low-cost $CO_2$ sensors introduces challenges regarding accuracy and inconsistent behavior between instruments that often requires labor-intensive regular calibration, data correction and filtering, and validation with

comparison to a smaller number of frequently calibrated high-accuracy instruments. In particular, the low-cost NDIR absorption sensor used in each BEACO$_2$N node (Vaisala CarboCap GMP343) is susceptible to temporal drift and fluctuations due to environmental variables that present challenges to achieving a goal of 1 ppm network error (van Leeuwen, 2010; Shusterman et al., 2016). Correction of the Vaisala CarboCap GMP343 instruments (Vaisala, hereafter) for changes in pressure, temperature, and humidity is required for accurate measurements (Vaisala, 2013). The typical correction for pressure and temperature

accounts for changes in the number density of $CO_2$ according to the ideal gas law (van Leeuwen, 2010; Vaisala, 2013; Shusterman et al., 2016). The humidity effect on measured $CO_2$ is accounted for by considering the dilution effect of water vapor according to Dalton's law of partial pressures (van Leeuwen, 2010; Vaisala, 2013; Shusterman et al., 2016). However, even after accounting for these factors, reported corrected $CO_2$ concentrations for the Vaisala instrument have been observed to exhibit a strong temperature dependence of up to 1 ppm/°C (van Leeuwen, 2010). Using a laboratory calibration procedure, van

Leeuwen (2010) found that a linear correction was necessary to account for the residual temperature dependence. However, correcting for the temperature dependence using lab calibrations is labor intensive, as the temperature dependence is unique for each Vaisala sensor. Regular laboratory temperature calibration would also be required to account for temporal variations in the temperature correction as sensors age. For a high-density urban network like BEACO$_2$N, this would require substantial time investment by trained personnel. The associated high labor costs defeat the purpose of using low-cost sensors. *In situ* field

calibration of the Vaisala sensors thus presents a more attractive method for correcting for the temperature dependence of the $CO_2$ measurements.

## 2.1 BEACO$_2$N network

The Berkeley Environmental Air-quality and $CO_2$ Network (BEACO$_2$N) Bay Area deployment currently consists of 73 nodes spaced at ∼2 km intervals with locations in Alameda, San Francisco, Contra Costa, Sonoma, Sacramento, and Solano counties.



A full description of a BEACO$_2$N node can be found in Kim et al. (2018). Briefly, each node contains a non-dispersive infrared
Vaisala CarboCap GMP343 CO$_2$ sensor, along with a Shinyei PPD42NS nephelometric particulate matter sensor and several
Alphasense electrochemical sensors for measuring CO, NO, NO$_2$, and O$_3$ (CO-B4, NO-B4, either NO2-B42F or NO2-B43F,
and either Ox-B421 or Ox-B431). The most recent version adds a Plantower PMS 5003 aerosol sensor. Sensors are assembled
into compact, weatherproof enclosures with air flow through the enclosure provided by two 30 mm fans. Data is compiled with
a Raspberry Pi microprocessor and an Adafruit Metro Mini microcontroller. Data is acquired every 5 or 10 s and is transferred
to a central server via an Ethernet or Wi-Fi connection. Observations are posted on the BEACO$_2$N website within a few hours
of measurement time (http://beacon.berkeley.edu).

The calibration procedure for the Vaisala CarboCap GMP343 CO$_2$ sensor is as outlined in Shusterman et al. (2016, 2018).
Briefly, deployed Vaisala sensors operate with the internal relative humidity (RH), temperature, and pressure compensation set
to "off" and the oxygen correction set to "on", with oxygen input as 20.95%. A post hoc multiplicative scale factor is applied
to convert the raw CO$_2$ outputs to the mole fraction of CO$_2$ that would be measured if the observed air parcel were dried and
brought to standard temperature and pressure ([CO$_2$]$_{STP}$). Raw CO$_2$ data is adjusted using temperature ($T$) measured by the
internal thermometer of the Vaisala. Water vapor pressure (P$_{H_2O}$) and air pressure (P$_{tot}$) are obtained from the pressure and
dew point temperature measured inside each node enclosure by a Bosch Sensortec Adafruit BME280 sensor. The [CO$_2$]$_{STP}$
is then adjusted to account for temporal drift in the instrument "zero" by comparing the background signal of the Vaisala
CO$_2$ measurement at each node to a reference Picarro G2301 system, located at the Richmond Field Station in Richmond,
CA (Fig. 1). A moving 3-week window of the 10$^{th}$ percentile of Vaisala CarboCap CO$_2$ data $\left(^{Vaisala}[\text{CO}_2]_{10\%}\right)$ is gener-
ated and compared with the 10$^{th}$ percentile of the reference Picarro instrument $\left(^{Picarro}[\text{CO}_2]_{10\%}\right)$. The difference between
$^{Vaisala}[\text{CO}_2]_{10\%}$ and $^{Picarro}[\text{CO}_2]_{10\%}$ is used to define the offset of the Vaisala instrument $\left(\left([\text{CO}_2]_{offset}^{T,drift}\right)\right)$. A linear cor-
relation between $\left([\text{CO}_2]_{offset}^{T,drift}\right)$ and time is generated and used to calculate the drift-corrected CO$_2$ data, $\left([\text{CO}_2]_{corrected}^{drift}\right)$
(Eq. 1—2).

$$\left([\text{CO}_2]_{offset}^{T,drift}\right) = m_t \times \text{days} + b \tag{1}$$

$$[\text{CO}_2]_{corrected}^{drift} = [\text{CO}_2]_{STP} - m_t - b \tag{2}$$

where $m_t$ is the temporal drift (ppm day$^{-1}$]) and $b$ is a constant atemporal bias.

## 2.2   Picarro reference instrument

A "supersite" with reference grade instruments is operated within the BEACO$_2$N Bay Area network to provide a reference
for the network calibration. Instruments are installed within a temperature-controlled instrument shelter at the U.C. Berke-
ley Richmond Field Station. Measurements include basic meterology, NO$_x$ (Thermo 42CTL with a molybedum NO$_2$ to NO
convertor), O$_3$ (Teledyne/API T400), CO$_2$, CH$_4$, and CO (Picarro G2401 cavity ring down analyzer). Air is sampled through



Teflon tubes mounted to a small tower affixed to the trailer roof, for a combined height of 6 meters above the ground. The co-located BEACO$_2$N node is attached outside of the trailer to the same tower.

The NO$_x$ and Picarro analyzer calibrations are checked against reference gases every two to three weeks. The reference gas cylinders for NO$_x$, CO, and CH$_4$ are Certified Standard grade from Praxair, and for CO$_2$ are from the NOAA Global Monitoring

Laboratory (two levels: 403.61 and 687.47 ppm). The Picarro checks are made by flowing the sequence of references gases into a tee at the inlet of the instrument for 15 minutes per step. The sequence of steps is performed twice during a check. The flow rate is set to be larger than the instrument sample flow (0.4 liters/minute) to overflow the inlet. The O$_3$ analyzer is checked against a photometric calibrator (Teledyne/API 703E).

## 2.3 Identification of a temperature-dependent error in Vaisala measurements

There exists an additional temperature dependence among the Vaisala CarboCap GMP343 instruments that varies between instruments. The temperature dependence was first identified from observations of CO$_2$ diurnal cycles at certain Bay Area BEACO$_2$N sites that were out of phase or larger in magnitude than the diurnal cycles at near-by nodes or measured by the Picarro. The presence of a temperature dependence in suspect Vaisala instruments was confirmed by examining the relationship between temperature in the node and the difference between baseline CO$_2$ signals measured by the Vaisala and the Picarro

reference instrument.

Diurnal cycles of urban CO$_2$ typically exhibit a daily maximum at night or mid-morning (depending on influence from traffic emissions) due to mixing in a shallow nighttime planetary boundary layer (PBL), and reach a minimum during the day as PBL height increases and vegetation takes up CO$_2$ (Idso et al., 2002; Coutts et al., 2007; Turnbull et al., 2015; Shusterman et al., 2016). The presence of an additional temperature dependence the Vaisala CO$_2$ instrument is particularly pronounced and

obvious in the measurements obtained with the sensor located at the East Bay Municipal Utility District (EBMUD) BEACO$_2$N site during 2020 (Fig. 2). The magnitude of the diurnal cycle at EBMUD is larger and out of phase with the Picarro reference instrument (Fig. 2a). The result of this temperature dependence at EBMUD (Fig. 2c) is a diurnal cycle that peaks midday (Fig. 2b). Figure 2b compares the CO$_2$ diurnal cycle at EBMUD with the nearby urban site Laney College (Fig. 1). In contrast to EBMUD, Laney College exhibits a daily maximum in the mid-morning–a pattern more consistent with typical urban CO$_2$

behavior (Idso et al., 2002; Coutts et al., 2007; Turnbull et al., 2015; Shusterman et al., 2016).

The Vaisala temperature dependence varies in magnitude and sign. Figure 3 shows the CO$_2$ mixing ratios and temperature dependence at the Montclair Elementary School site. Compared to the Picarro instrument, this site also demonstrates higher amplitude diurnal cycles (Fig. 3a), but these diurnal cycles are in phase with the reference instrument. Unlike EBMUD, the Montclair site exhibits a negative temperature dependence (Fig. 3c). Figure 3b shows the diurnal cycles at Montclair and the

nearby node located at College Preparatory School (CPS). The comparison of these two sites suggests there may indeed be an amplification of the diurnal cycle at Montclair caused by a negative temperature dependence of the Vaisala instrument.



## 2.4 Temperature correction method

The goal of our approach to accounting for temperature dependence of the Vaisala instruments is to rely exclusively on the network itself and, if available, supplementary reference instruments, such as a Picarro, for derivation of correction factors to null sensor temperature dependence.

The method we developed builds on our method for accounting for drift in the instrument zero. To derive a temperature factor, we use hourly averaged $[CO_2]_{STP}$ and node measurements of temperature ($T$). It is important to note that a major factor contributing to the temperature inside the node is whether the node is placed in the sun or shade. As a result, direct correlation with meteorological temperature measured outside the node is not strong. For a moving three week window, at each hour ($h$), the lowest 10th percentile of $[CO_2]_{STP}$ within $\pm\ 1$ °C of $T(h)$ is calculated. A running array of temperature-based 10th percentile data is created for both the Picarro supersite $\left(^{Picarro}[CO_2]_{10\%}^{T}\right)$ at the Richmond Field Station and each Vaisala instrument $\left(^{Vaisala}[CO_2]_{10\%}^{T}\right)$ using the temperature ($T$) of the Vaisala instrument. The Vaisala temperature is assumed to be the temperature that the instrument is responding to. $\left(^{\Delta}[CO_2]_{10\%}^{T}\right)$ is then calculated, where:

$$^{\Delta}[CO_2]_{10\%}^{T} =^{Vaisala} [CO_2]_{10\%}^{T} -^{Picarro} [CO_2]_{10\%}^{T} \tag{3}$$

A linear regression for $^{\Delta}[CO_2]_{10\%}^{T}$ against $T$ provides a slope ($m_T$) and intercept ($b_T$) for a moving three-week time window. We considered the possibility that the instrument response to temperature could be a zero shift and/or a change in the response to $CO_2$. We were able to achieve similar results assuming the temperature effect is entirely due to one or the other of these possibilities. As there is already substantial drift in the instrument zero, we proceed under the assumption that the effect can be entirely attributed to the temperature dependence of the instrument zero. The median of $m_T$ is calculated for the deployment period of the Vaisala sensor to determine the temperature-corrected offset and $CO_2$ mixing ratios of Vaisala $CO_2$ measurements, based on an additive error correction (Eq 4—5). When it is observed the either the offset bias, the temperature-dependent slope, or the time-dependent drift in the instrument zero shifts dramatically during a deployment period, the deployment is manually separated into different periods that are calibrated separately.

$$[CO_2]_{offset}^{T} =^{\Delta} [CO_2]_{10\%}^{T} -^{med} m_T \times T \tag{4}$$

$$[CO_2]_{corrected}^{T} = [CO_2]_{STP} -^{med} m_T \times T \tag{5}$$

An example calibration, demonstrating $m_T$ and $[CO_2]_{offset}^{T}$ over time at EBMUD 2020, is shown in Figure S1. Following calculation of the temperature-corrected offset, the temporal drift slope and intercept of this corrected offset are calculated and corrected using the methods described above, resulting in the generation of the temperature- and drift-corrected $CO_2$ offset $\left([CO_2]_{offset}^{T,drift}\right)$.





The final temperature- and drift-corrected $CO_2$ $\left([CO_2]_{corrected}^{T,drift}\right)$ is then caluclated as:

$$[CO_2]_{corrected}^{T,drift} = [CO_2]_{STP} - ^{med}m_T \times T - [CO_2]_{offset}^{T,drift} \qquad (6)$$

The majority of the BEACO$_2$N nodes examined demonstrated a strong linear relationship between $\left(^{\Delta}[CO_2]_{10\%}^T\right)$ and node temperature. However, the node at Elsa Widenmann Elementary School appeared to show a strong negative temperature dependence only on particularly warm days (Fig. 4a,c). The temperature dependence of $\left(^{\Delta}[CO_2]_{10\%}^T\right)$ for this node better fit

a quadratic than an linear relationship. To account for nodes with a non-linear temperature dependence, in cases where a quadratic fit improves the $R^2$ of the fit by more than 0.2, the $\left([CO_2]_{offset}^T\right)$ and $\left([CO_2]_{corrected}^{T,drift}\right)$ are calculated via Eq.7—8.

$$[CO_2]_{offset}^T = ^{\Delta}[CO_2]_{10\%}^T - ^{med}m_T^1 \times T - ^{med}m_T^2 \times T^2 \qquad (7)$$

$$[CO_2]_{corrected}^{T,drift} = [CO_2]_{STP} - ^{med}m_T^1 \times T - ^{med}m_T^2 \times T^2 - [CO_2]_{offset}^{T,drift} \qquad (8)$$

$m_T^1$ and $m_T^2$ are the first and second terms of the quadratic fit of $^{\Delta}[CO_2]_{10\%}^T$ against $T$.

We attempted to determine a relationship between Vaisala sensor age and temperature-dependence slope, but $m_T$ was only weakly correlated with sensor age ($r \approx 0.3$). We did, however find some evidence that older sensors had a larger likelihood of having a larger temperature-dependence. For sensors less than 3 years since their initial deployment, 90% had $m_T < 1$ ppm/°C and 64% had $m_T < 0.5$ ppm/°C. For sensors older than 3 years, 75% had $m_T < 1$ and 47% had $m_T < 0.5$.

## 3   Evaluation of calibration

Figures 5b, 5e, and 4c show the temperature dependence of $^{\Delta}[CO_2]_{10\%}^T$ nodes located at at EBMUD, Montclair, and Elsa Widenmann, respectively. Figures 5a, 5d, and 4b show a comparison of the data at EBMUD, Montclair, and Elsa, respectively, with and without and adjustment for a temperature-dependent zero offset. With the application of the temperature correction, the magnitudes of the diurnal cycles are reduced and demonstrate much better agreement in amplitude and phase with the Picarro instrument. The resulting diurnal cycle at EMBUD shows a much more typical diurnal cycle for an urban site, with

a maximum occuring in the mid-morning (Fig. 5c). At Montclair, the magnitude of the diurnal cycle is reduced, reaching a maximum of $\sim 430$ ppm $CO_2$ during the early morning, and a minimum of $\sim 412$ ppm $CO_2$ during midday–a pattern much more aligned with the diurnal cycle exhibited at CPS (Fig. 3b, Fig. 5c).

Following confirmation of the effectiveness of the temperature correction method on the sensor deployed at EBMUD in 2020 (EBMUD 2020, hereafter sensors will be referred to following the notation: site year) and Montclair 2018, we examined

the temperature-corrected $CO_2$ data at the Laney College BEACO$_2$N site during the spring (March—June) of three different years when different Vaisala CarboCap GMP343 instruments were deployed at the site. Given the hypothesis that the observed


temperature dependence is due to temperature-dependent errors in the Vaisala $CO_2$ signal, a successful calibration should be sensor specific, rather than site specific. Figure 6b demonstrates the different $^{\Delta}[CO_2]^T_{10\%}$ temperature dependence during three different years with different instrument deployments. Each deployment has a distinct offset and slope of $^{\Delta}[CO_2]^T_{10\%}$ vs tem-

perature. During all deployment years, the temperature correction results in better agreement between the reference instruments and the Vaisala data (e.g. 6/15/2018), while preserving local signals (e.g. 4/14/2020) (Fig: 6a). The correction is also effective for the data record before deployment of the Picarro reference instrument in August 2017, when the Exploratorium $CO_2$ Buoy, located in the San Francisco Bay, was used as a reference instrument (Fig. 6a). The correction of the $CO_2$ diurnal cycle at Laney College is most notable during 2017, although midday levels of $CO_2$ are reduced in the corrected data for 2018 and

2020 as well (Fig 6c).

The temperature correction method was further validated by examining neighboring sites in two regions of the Bay Area during and before periods of high $CO_2$ during September 2020 northern California fires. The Richmond sites of Washington Elementary School, Nystrom Elementary School, Dejean Middle School, and Peres Elementary and the Vallejo sites of Beverly Hills Elementary School, Mare Island Health and Fitness Academy, Grace Patterson Elementary School, and Highland

Elementary School were compared. The resulting temperature dependent percent differences of $CO_2$ between adjacent sites are reduced to approximately 0–2% from 1–5% (Fig. S3, S6). Temperature corrections also result in better agreement in $CO_2$ mixing ratios between adjacent sites in Richmond (Fig. 7 and Fig. S2) and in Vallejo (Fig. S4, S5). The results were identical when a multiplicative correction term, rather than additive, was considered (e.g. if the temperature effect was assumed to be on the $CO_2$ signal magnitude rather than entirely on the instrument zero).

## 3.1 Comparison of nearest-neighbor sites

To assess the improvement in the network precision following application of the temperature-dependence correction, we combined observations from the entire Bay Area network using data from all of 2020. All sites with available data for more than one month of 2020 were included. Nearest neighbor pairs of each site were identified, where nearest neighbors to an individual site were considered as the closest BEACO$_2$N sites within a 2 km radius of the site. There are 53 unique nearest neighbor pairs.

For each nearest neighbor pair $X$ and $Y$, an array of the fractional differences between sites were calculated as: $([CO_2]_X - [CO_2]_Y)/[CO_2]_X$. This was done using both the measurements before and after correction for temperature-dependent instrument zero $\left([CO_2]^{drift}_{corrected} \text{ and } [CO_2]^{T,drift}_{corrected}\right)$. Figures 8a and 8d show the fractional differences of each nearest neighbor pair as a histogram calculated using $[CO_2]^{drift}_{corrected}$ and $[CO_2]^{T,drift}_{corrected}$, respectively. Most nearest neighbor site pairs exhibit a distribution of fractional differences centered close to zero, with both positive and negative tails (Fig. 8a,d). The tempera-

ture correction results in a clear improvement of agreement between nearest neighboring sites, with the mean of the absolute value of the average fractional differences of all nearest neighbor pairs decreasing by a factor of 2 from 0.025 to 0.013. For $[CO_2]^{T,drift}_{corrected}$, this represents an average difference of 6.5 ppm at $[CO_2]$ = 500 ppm. Figures 8b and 8e express the fractional differences of nearest neighbor pairs as a single distribution calculated using $[CO_2]^{drift}_{corrected}$ and $[CO_2]^{T,drift}_{corrected}$, respectively. Fit to a Lorentz distribution, the mean and scale parameter of the distribution of nearest neighbor pairs using $[CO_2]^{drift}_{corrected}$

is 0.0026 and 0.014, respectively, without accounting for temperature dependence and there is a substantial narrowing of the





distribution, resulting in a mean and scale parameter of 0.005 and 0.007, respectively, after accounting for the effect of a temperature dependent offset.

Further analysis was performed to confirm that the temperature correction method eliminates any temperature-dependent disagreement between nearest neighboring sites. The nearest neighbor fractional differences of $CO_2$ data were separated into 2 °C temperature bins. For each temperature bin, the absolute value of the mean fractional difference between each nearest neighbor pair, using either $[CO_2]_{corrected}^{drift}$ or $[CO_2]_{corrected}^{T,drift}$, was calculated. We then averaged the mean fractional difference in each temperature bin over all nearest neighbor pairs. A plot of the resulting network mean percent difference vs. temperature is shown in Figures 8c and 8f, using $[CO_2]_{corrected}^{drift}$ and $[CO_2]_{corrected}^{T,drift}$ data, respectively. In the original data, the mean percent differences were greatest at both high and low temperatures. In the temperature-corrected data, there is no clear dependence of nearest neighbor mean percent differences on temperature. The mean percent difference at all temperatures is also reduced.

## 4 Assessment of the network error

Turner et al. (2016) suggested that a network uncertainty of ~1 ppm $CO_2$ would be compatible with relevant constraints on point, line, and area $CO_2$ sources of 147, 45, and 9 tC hr$^{-1}$, respectively. Assessing network error in the field is, however, a complex problem. We approach the problem by exploring differences between adjacent nodes, which should be an upper limit to the uncertainty. Although the site-to-site variation is strongly influenced by local emissions sources, there are also strong correlations with changes in urban-, synoptic-, and global-scale $CO_2$ signals that are spatially coherent across pairs of adjacent nodes. Variances between adjacent nodes are due to a combination of true site-specific signals and instrument biases. It is therefore difficult to know the minimum variance in adjacent nodes for a hypothetical "perfect" measurement. For nearest neighbor sites, the majority of the $CO_2$ signal should show near zero difference, representing the background signal. In the observation record we would also expect moments when either site in a pair has a larger signal, driven by local emission sources and meteorology. Sites closer to the highway also typically have larger $CO_2$ signals (Shusterman et al., 2018). In the following section we describe a procedure for evaluating network error and summarize the improvements following inclusion of the temperature-correction described above.

### 4.1 Site variance and correlation lengthscales

To evaluate the network error, a semivariogram was constructed for $[CO_2]_{corrected}^{T,drift}$ (Fig. 9). Using data from all sites with more than three days of available data during the summer of 2020, we calculated the semivariance between $CO_2$ measurements at each BEACO$_2$N node and all other sites in the Bay Area network. Summer months were chosen because the average and diurnal variability of $CO_2$ mixing ratios are reduced, meaning that measured site variances are relatively more influenced by instrument error, rather than by "true" atmospheric variance, than in the winter. In Figure 9 the square root of the semivariance is plotted against the distance separating the BEACO$_2$N nodes and fitted with an exponential model. The Picarro reference instrument at the Richmond Field Station was included in this analysis.



Using the root semivariance as a correlation metric, in temperature-corrected data, the $e$-folding length scale for variation is $1.2 \pm 0.3$ km ($1.7$ km $\pm 0.7$ km using semivariance as a correlation metric, not shown), supporting the BEACO$_2$N hypothesis that 2 km node spacing in a dense network will capture important elements of local variability. The temperature-correction

results in a maximum root semivariance of $5.5 \pm 2$ ppm (reduced from 8 ppm in the uncorrected data). Extrapolated to a distance of zero, the temperature correction method has a predicted root semivariance of $1.3 \pm 0.9$ ppm, representing the network error. This analysis suggests that the desired $\sim 1$ ppm network error has been achieved with the application of the temperature-correction.

Length scales for correlations ($r^2$) between sites calculated by Shusterman et al. (2018) during the summer 2017 were larger

than the 1.2 km length scale identified here for root semivariance (1.7 km for semivariance). To more directly compare, we also performed the method of Shusterman et al. (2018) on the temperature-corrected CO$_2$ data for the summer of 2020. We examined the correlation of CO$_2$ concentrations for every pairing of Bay Area sites during this period for all hours, during the day, and during the night (Fig. S7). The $e$-folding distance for the decay of $r^2$ correlation coefficients was 2.8 km for all times, 3.7 km during the day, and 2.8 km at night. This is in good agreement with the length scales of 2.9 km at all times, 3.6 km

during the day, and 2.2 km at night found by Shusterman et al. (2018). The base-line correlation for sites separated by more than 20 km was found to be 0.46, larger than the correlation background of $\sim 0.3$ of Shusterman et al. (2018). The temperature correction does not affect the characteristic length scale of BEACO$_2$N sites, but improves the overall base-line correlations and variances.

## 4.2    Contribution of instrument error to site variance

We can represent the network instrument error also by examining the sources contributing to the semivariance between nearest neighboring sites. The semivariance ($\gamma_{nn}$) of nearest neighboring sites can be expected to have contributions from both "true" variations in emissions and meteorology and erroneous differences caused by instrument error. To estimate the portion of the semivariance resulting from atmospheric phenomenon, an analogous quantity for the hourly variations in CO$_2$ was calculated for each site according to Eq. 9:

$$\gamma_{hh} = \frac{1}{2N} \sum_{1}^{N-1} ([CO_2]_h - [CO_2]_{h+1})^2 \tag{9}$$

$N$ is the number of hours of data and $[CO_2]_h$ , and $[CO_2]_{h+1}$ are the measured mixing ratios of CO$_2$ at each hour and one hour later, respectively. The individual instrument error was then calculated as:

$$\epsilon_{inst} = \sqrt{\gamma_{nn} - \gamma_{hh}} \tag{10}$$

The resulting upper-bound instrument error from the median of individual instrument errors for the Bay Area network is $2.5 \pm$

$0.5$ ppm. (This estimate for non-temperature corrected data is $4.5 \pm 0.9$ ppm). We consider this an upper bound because hourly variations in the CO$_2$ signal reflects the atmospheric changes at an individual site, which may not match with the atmospheric





changes at the nearest neighbor sites. Variations in emissions or wind velocity, may result is larger "true" differences between a site and its nearest neighbor than are represented by the site's hourly variability.

To reduce the influence from short-term atmospheric variations, the network error was also estimated using an individual site's root mean squared error ($RMSE_i$) as a metric for "true" atmospheric variation (Eq. 11) and a "paired" RMSE ($RMSE_{paired}$) using the mean $CO_2$ signal of its nearest neighbor site ($\overline{nn[CO_2]}$) as a measure of total variation (Eq. 12). The site error was then calculated according to Eq. 13

$$RMSE_i = \sqrt{\frac{\sum_N([CO_2]_h - \overline{[CO_2]_i}}{N}} \tag{11}$$

$$RMSE_{paired} = \sqrt{\frac{\sum_N([CO_2]_h - \overline{nn[CO_2]}}{N}} \tag{12}$$

$$\epsilon_{inst} = \sqrt{RMSE^2_{paired} - RMSE^2_i} \tag{13}$$

The resulting network instrument errors were between 0.5 ppm and 4 ppm, with a median of $1.6 \pm 0.4$ ppm, in good agreement with the error calculated from the semivariogram fit. Based on these analyses, we estimate the network error of the Bay Area BEACO$_2$N network to be less that 1.6 ppm, close to our stated goal of 1 ppm network error.

## 5 Application to other city networks

The BEACO$_2$N network has recently been extended to several other cities, and will further expand to additional locations in coming years. Currently, locations where BEACO$_2$N nodes are deployed (in addition to the Bay Area) are Houston (19 nodes, network start 11/2017), Glasgow in collaboration with the University of Strathclyde (>20 nodes, network start 5/2021), New York City (8 nodes, network start 4/2018), and Los Angeles, in collaboration with the University of Southern California (12 nodes, network start 5/2021). The goal of the network is to be self-calibrated, as not all locations at which the nodes will be deployed have a highly precise and frequently calibrated reference instrument. As such, an alternative method of obtaining a reference for the determination of drift, offset, and temperature dependence is needed.

We find that the network median $[CO_2]_{STP}$ ($[CO_2]^{med}_{STP}$) can be used as a reference. To begin, we define the network median ($[CO_2]^{med}_{STP}$) as the median $[CO_2]_{STP}$ of sites having a temperature dependent slope ($m_T$) less than 1 ppm/°C. $[CO_2]^{med}_{STP}$ is used as a "reference site" from which a temperature-based 10$^\text{th}$ percentile data ($^{med}[CO_2]^T_{10\%}$) is calculated for determination of $^\Delta[CO_2]^T_{10\%}$:

$$^\Delta[CO_2]^T_{10\%} = {}^{Vaisala}[CO_2]^T_{10\%} - {}^{med}[CO_2]^T_{10\%} \tag{14}$$





### 5.1 Bay Area Tests

We observe good agreement between the Picarro reference instrument during 2020 and $[CO_2]_{STP}^{med}$ (Fig. 10). The mean percent difference considering all 2020 data is 0.46%, representing an accuracy error of 2 ppm at 420 ppm $CO_2$ (Fig. 10d). We also do not see evidence of a temperature-dependent offset between the Picarro reference instrument and $[CO_2]_{STP}^{med}$.

The precision of the Bay Area network is negligibly affected when the network median is used as the reference, with the mean of the absolute value of the average fractional differences of all nearest neighbor pairs equal to $0.015 \pm 0.008$

(compared to $0.013 \pm 0.007$ with the Picarro as reference) (Fig. S8). The resulting maximum root semivariance is $5.5 \pm 2$ ppm and extrapolated root semivariance at zero km separation is $0.8 \pm 0.9$ ppm, respectively, approximately equal to the values calculated when the Picarro is used as a reference. The network accuracy is however, more appreciably altered. Figure 11 shows the fractional difference between $[CO_2]_{corrected}^{T,drift}$ determined using the Picarro and $[CO_2]_{STP}^{med}$ as a reference at each site. The resulting mean percent difference is $0.51 \pm 0.02$ %, representing a network accuracy error of 2 ppm at 420 ppm $CO_2$.

This accuracy error is mainly driven by small differences in the offsets (2 ppm on average) and $m_T$ (0.2 ppm/°C on average, see Supporting Information) between $[CO_2]_{corrected}^{T,drift}$ calculated using the Picarro and $[CO_2]_{STP}^{med}$ as a reference. These results suggest that the network precision can be expected to remain near 1 ppm $CO_2$ with the use of $[CO_2]_{STP}^{med}$ as a reference, but additional accuracy error of 2 ppm may be introduced.

Analysis of the Bay Area network was performed on the 36 nodes with sufficient data availability for 2020. However, the

newly established networks have fewer nodes than in the Bay Area. To use $[CO_2]_{STP}^{med}$ as a reference, we must have sufficient nodes from which to calculate the network median. To evaluate this, for $n = 1$—26, a random subset of $n$ Bay Area nodes was selected 100 times. For each of the 100 random subsets of $n$ nodes, the mean fraction difference was calculated between the network median $CO_2$ and the median calculated using the subset. The average and standard error of the 100 mean fraction differences was then calculated. The results of this analysis are presented in Figure 12. We suggest that a minimum of 7 nodes

with $m_T$ less than 1 ppm/°C is required for the accuracy error to be lower than 2%. For less than 1% error, at least 12 nodes are required.

### 5.2 Houston

Data from the Houston network was subsequently calibrated using $[CO_2]_{STP}^{med}$ as a reference for determination of temperature dependence, drift, and offset. Temperature dependence calibration of each site in the Houston network was performed twice. All

sites were first included in $[CO_2]_{STP}^{med}$ and sites with $m_T$ greater than 1 ppm/ °C were identified. These sites were then excluded from $[CO_2]_{STP}^{med}$ and each site was re-calibrated. Histograms of the fraction differences between nearest neighbor sites are shown in Figure 13. The average mean percent difference between nearest neighbors was $2 \pm 1$ %. Though considerably larger than the differences between nearest neighbors in the Bay Area network, it is not immediately clear whether this difference is caused by greater precision error in Houston, or differing meteorology and $CO_2$ sources that cause greater differences

between $CO_2$ mixing ratios at adjacent sites. We attempted to perform a similar instrumental error analysis, but there are



currently insufficient overlapping $CO_2$ data in Houston for uncertainty analysis. However, we do not have reason to expect the instrument errors would be any larger in the Houston network.

# 6 Conclusions

We have assessed the accuracy of the BEACO$_2$N network following *in situ* calibration of the temperature-dependence in Vaisala
$CO_2$ sensors. We report a network instrument error of 1.6 ppm $CO_2$ or less, near the desired 1 ppm network error suggested by Turner et al. (2016).

A method for correcting Vaisala instrument temperature dependence in BEACO$_2$N has been established and evaluated using sites across the San Francisco Bay Area network. The method corrects observations from individual instruments so that they exhibit a temperature dependence in their lowest temperature-based 10$^{th}$ percentile of $CO_2$ data that is equivalent to that of
a reference site, thus correcting erroneous instrument temperature dependence while preserving true diurnal cycles and local signals. This field calibration of temperature dependence can be entirely internal to the network and does not necessarily require a reference instrument, although the addition of a reference instrument provides greater network accuracy. The implementation of the temperature correction method produces more reasonable diurnal cycles, diurnal cycles that are maintained for sites influenced by similar emissions sources, and better agreement between adjacent sites. We additionally describe methods for
characterizing network scale uncertainties and site-to-site biases. The average variation between adjacent sites was found to be 1.3% following implementation the temperature correction (compared to 2.5% prior to the correction). The temperature correction greatly improves the precision of $CO_2$ measurements in the BEACO$_2$N network.

We show that the network precision can be maintained at 1.3% even in locations without a high-cost reference instrument, using the network median as a reference, provided that there are at least 12 sites with small temperature dependencies. This has
important implications for the expansion of BEACO$_2$N to additional cities globally, as well as for other dense low-cost $CO_2$ networks. However, without a reference instrument, the network accuracy error is larger and is $\sim \pm 2$ ppm.

*Author contributions.* JK, HLF, CN, and PW collected the data used in this analysis. ERD composed the manuscript and designed and executed the analysis in consultation with JK and KW. KW also aided with data processing and implementation of the temperature calibration method. JK and RCC provided additional manuscript feedback and RCC supervised the project.

*Competing interests.* The authors declare that they have no conflict of interest.

*Data availability.* The data used for this study is publicly available at http://beacon.berkeley.edu. Raw data can be given upon request.



*Acknowledgements.* This work was funded by the Koret Foundation and University of California, Berkeley. We also thank A. J. Turner for his input and former members of the BEACO$_2$N project for establishing the network: A. A. Shusterman, V. Teige, and K. Lieschke.



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

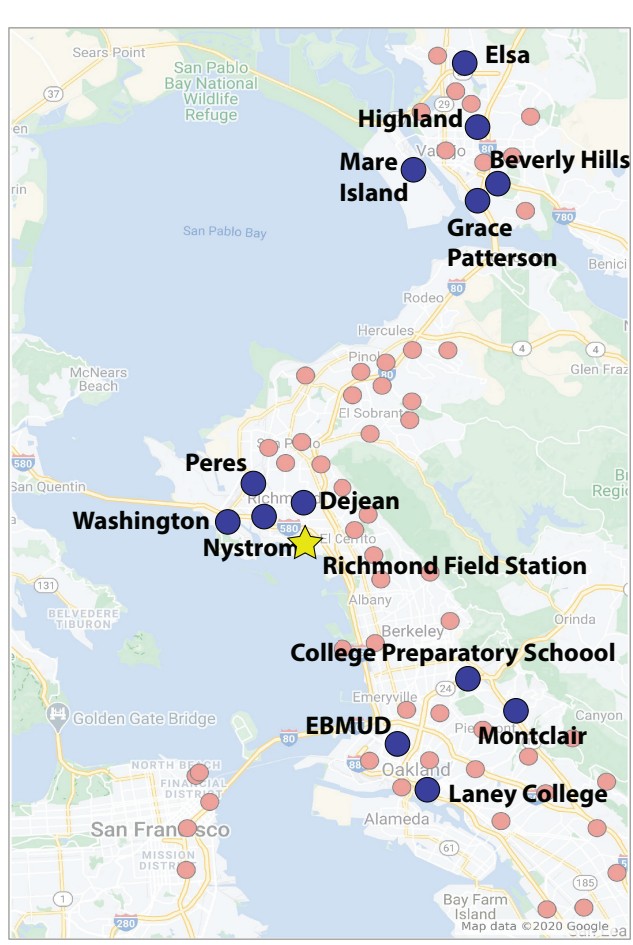

**Figure 1.** a) Map of all Bay Area BEACO$_2$N sites (small red dots), BEACO$_2$N sites discussed in this work (large blue dots), and the Richmond Field Station (star).



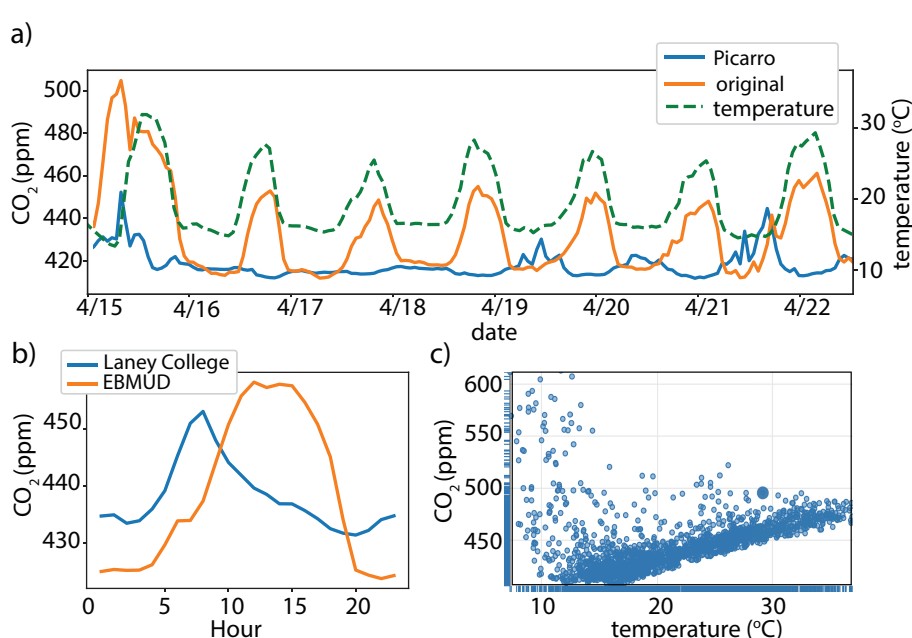

**Figure 2.** a) $CO_2$ mixing ratios from April 2020 at EBMUD and measured with a Picarro instrument at the Richmond Field Station supersite. b) EBMUD 2020 diurnal cycle compared with Laney College. c) Temperature dependence of the $CO_2$ signal at EBMUD.

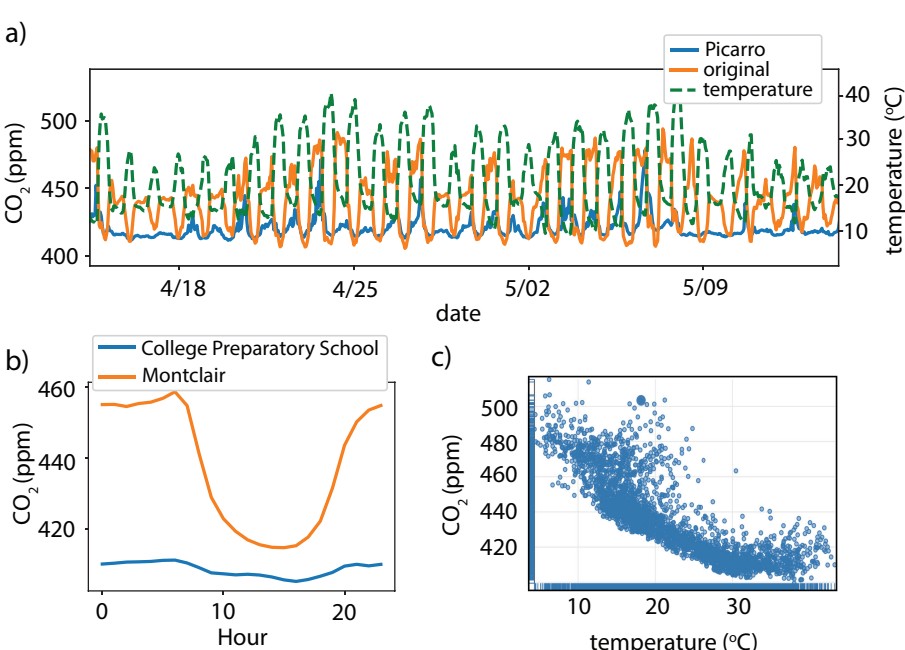

**Figure 3.** a) $CO_2$ mixing ratios from May 2018 at Montclair and measured with a Picarro instrument at the Richmond Field Station supersite. b) Montclair 2018 diurnal cycle compared with College Preparatory School. c) Temperature dependence of of $CO_2$ signal at Montclair.



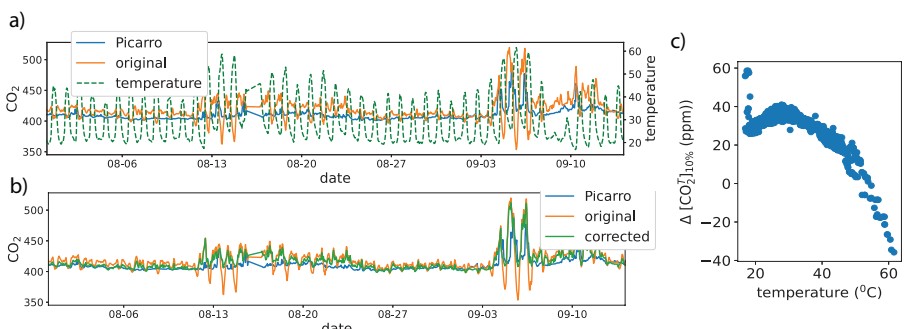

**Figure 4.** a) $CO_2$ mixing ratios measured by the Picarro instrument at the Richmond Field Station (blue solid), uncorrected $CO_2$ measured at Elsa Widenmann Elementary School (orange solid), and node temperature measured at Elsa (green dashed). (b) $CO_2$ mixing ratios at Elsa Widenmann Elementary School with no temperature correction (green), temperature correction applied (green) and measured with a Picarro instrument at the Richmond Field Station (blue). c) Temperature dependence at Elsa Widenmann Elementary School of $^{\Delta}[CO_2]_{10\%}^{T}$.

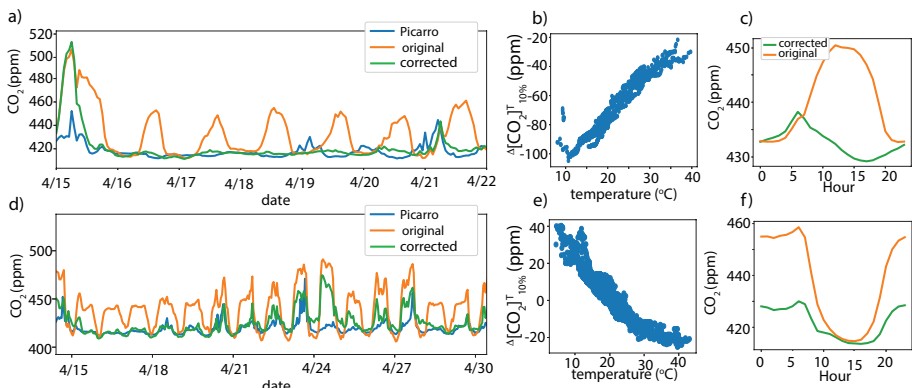

**Figure 5.** $CO_2$ mixing ratios at (a) EBMUD and (d) Montclair with no temperature correction (orange), temperature correction applied (green) and measured with a Picarro instrument at the Richmond Field Station supersite (blue). Temperature dependence of $^\Delta[CO_2]^T_{10\%}$ at (b) EBMUD and (e) Montclair. Diurnal cycle with and without temperature correction at (c) EBMUD and (f) Montclair.

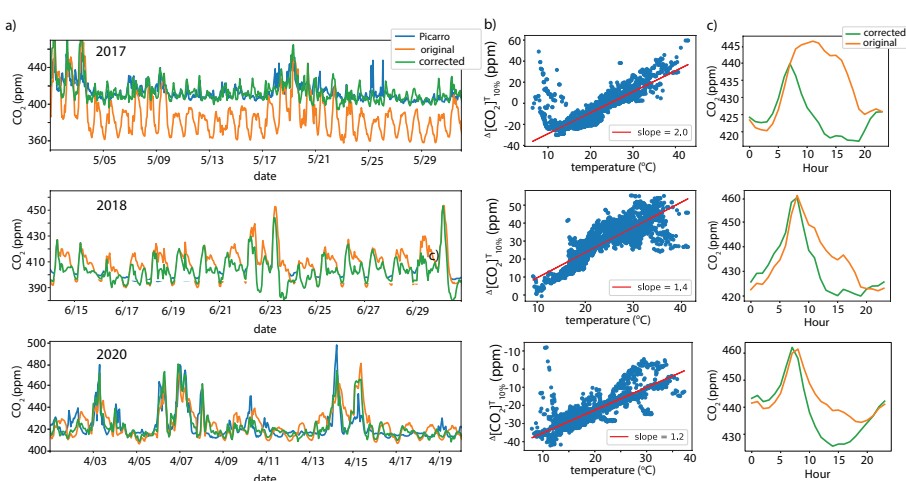

**Figure 6.** Data from 2017 are in top panels, 2018 are in middle panels, 2020 are in bottom panels. a) $CO_2$ mixing ratios at Laney College with no temperature correction (green), temperature correction applied (blue) and measured with a Picarro instrument at the Richmond Field Station supersite (2018 and 2020) or with the Exploratorium Buoy (2017). b) Temperature dependence at Laney College of $^{\Delta}[CO_2]^T_{10\%}$. c) Laney College diurnal $CO_2$ cycle with (green) and without (orange) temperature correction.



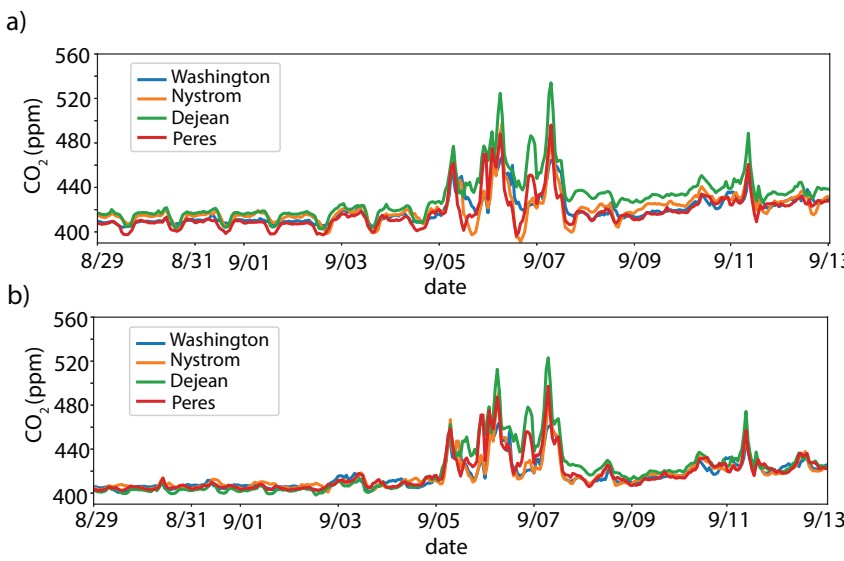

**Figure 7.** $CO_2$ mixing ratios during and before 2020 September wildfires at four adjacent sites in Richmond without (a) and with (b) temperature correction.



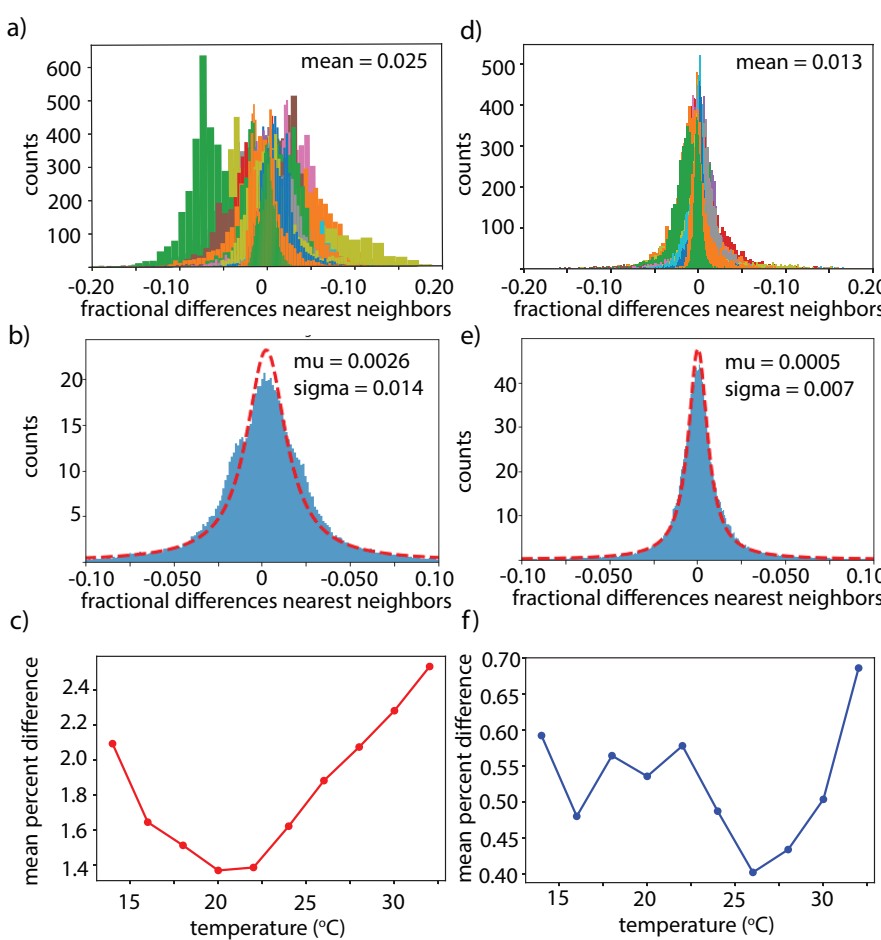

**Figure 8.** Histogram of the fractional differences between nearest neighbors sites a) without and d) with the temperature correction applied. Different colors represent different pairs of neighboring sites. Histogram of the fractional differences between all aggregated nearest neighbors sites b) without and e) with the temperature correction applied fit to a Lorentz distribution. Network mean of the percent difference for each nearest neighbor pair averaged by 2 °C bins c) without and d) with temperature correction applied.

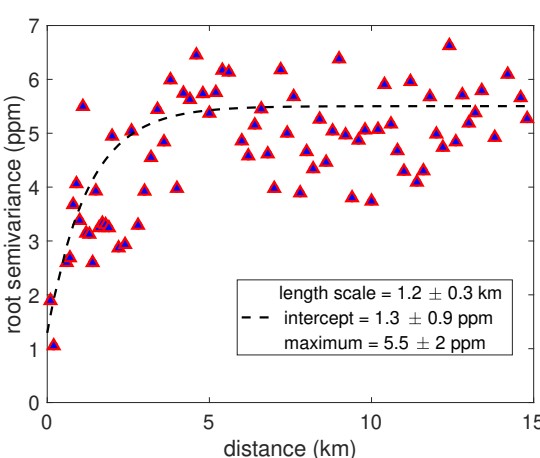

**Figure 9.** Semivariogram of BEACO$_2$N sites for data with temperature correction applied. Data are averaged by 0.1 km bins. Plot includes data from the Picarro instrument at Richmond Field Station.

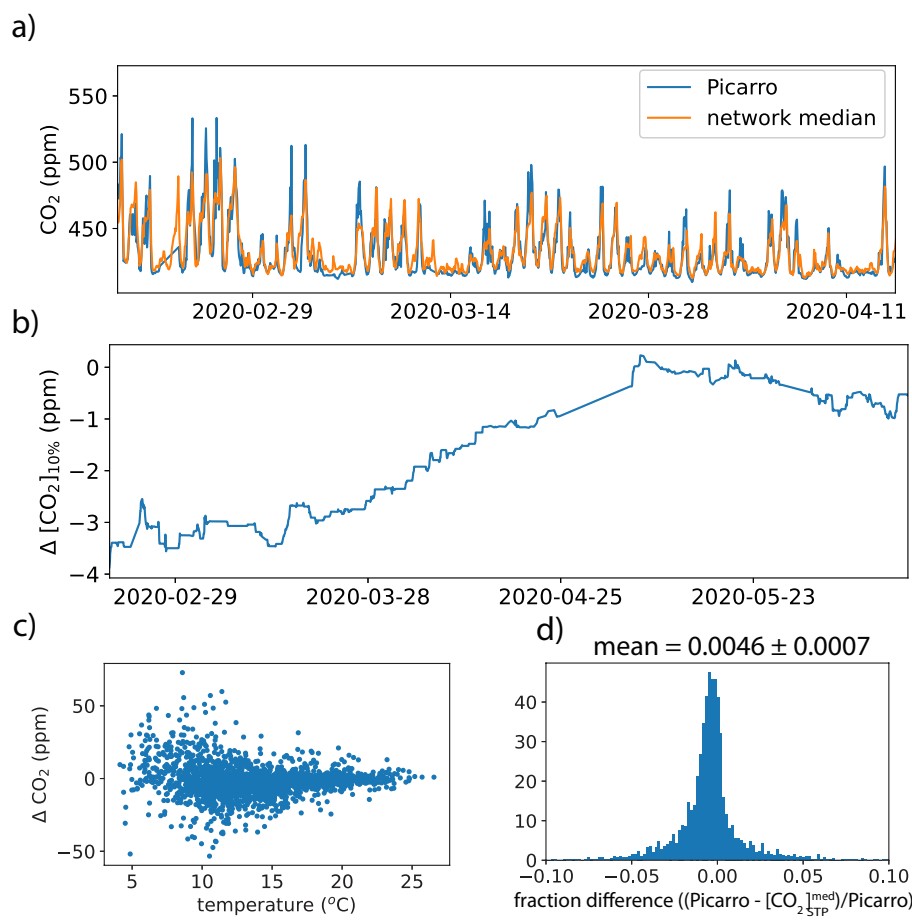

**Figure 10.** a) $CO_2$ mixing ratios measured by the Picarro instrument at the Richmond Field Station (blue) and the median $CO_2$ of all Bay Area nodes having a temperature dependent slope less than an absolute value of 1 ppm/°C (orange). b) The difference in the tenth percentile of $CO_2$ mixing ratios measured by the Picarro instrument and the network median plotted versus date and (c) versus temperature. (d) Histogram of the fractional differences between Picarro $CO_2$ mixing ratios and the network median. Data for (c) and (d) include all of 2020.

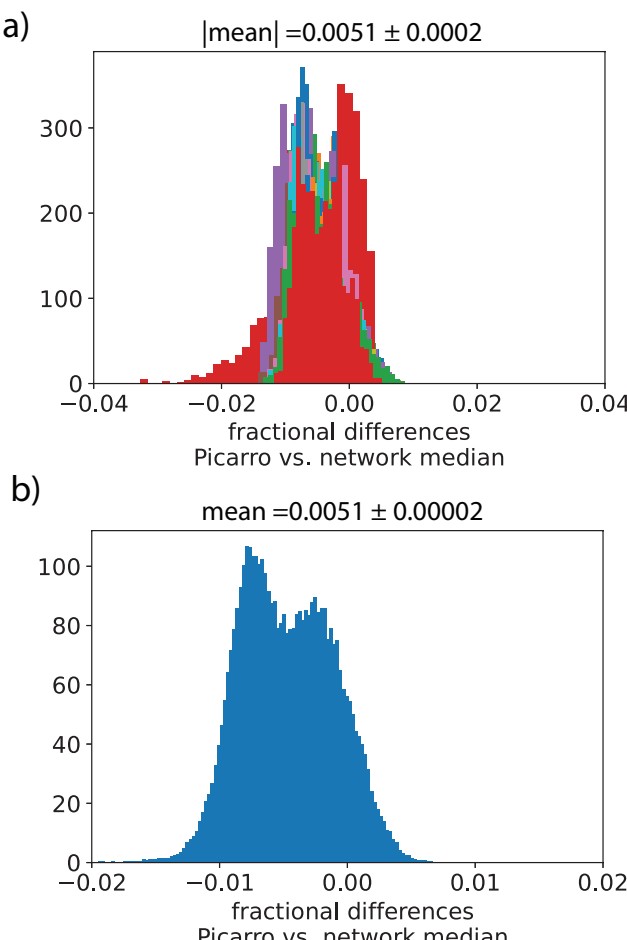

**Figure 11.** (a) Histogram of the fractional differences between sites with temperature-corrected $CO_2$ calculated using the Picarro instrument and the Bay Area network median as a reference. Different colors represent different sites. The mean indicated is the average of the absolute values of each neighboring pair's mean fractional difference. b) Histogram of the aggregated fractional differences between sites with temperature-corrected $CO_2$ calculated using the Picarro instrument and the Bay Area network median as a reference. The mean and error indicated is the mean and 95% confidence interval of the distribution.

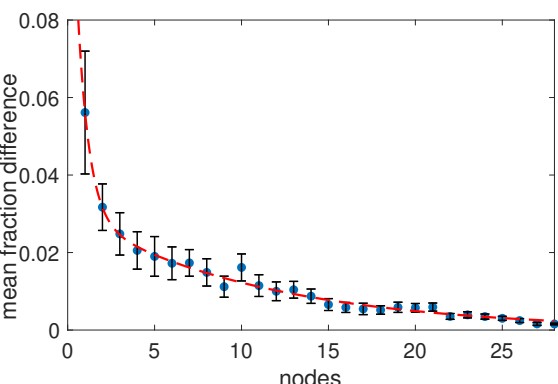

**Figure 12.** Fraction difference between the Bay Area network median calculated from all Bay Area sites and the network median calculated from a subset of between one and 26 nodes. A random subset of $n = 1$—26 nodes were selected to calculate the mean fraction difference between the network median $CO_2$ and the median calculated using the subset. This was repeated 100 times for each of $n = 1$—26 nodes. The reported fraction difference and error bars are the average and 95% confidence interval of the mean fraction difference from the 100 random samples.



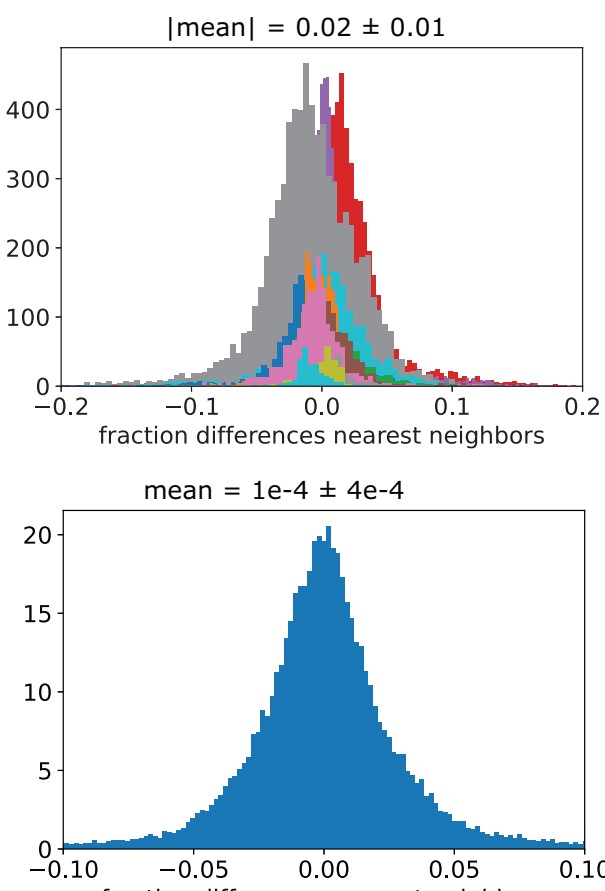

**Figure 13.** a) Histogram of the fractional differences between nearest neighbors sites in the Houston network with the temperature correction applied using the network median as a reference. Different colors represent different pairs of neighboring sites. The mean and error indicated is the average and 95% confidence interval of the absolute values of each neighboring pair's mean fractional difference. b) Histogram of the fractional differences between all aggregated nearest neighbors sites with the temperature correction applied. The mean and error indicated is the mean and 95% confidence interval, respectively, of the distribution.