# Peer review of "The Berkeley Environmental Air-quality and CO2 Network: field calibrations of sensor temperature dependence and assessment of network scale CO2 accuracy"

_Atmospheric Measurement Techniques, 2021_

## Author Response (AR1)

We kindly thank the two reviewers for their constructive and thoughtful comments. We have incorporated the reviewers concerns into the revised manuscript as indicated below and in the document with tracked changes. Reviewer comments below appear in **bold** text and our responses are in standard font.

**Response to Reviewer #1**

**Delaria et al. evaluate the performance of the low-cost sensor network BEACO₂N for $CO_2$ measurements in the Bay Area, California. They find that, the low-cost sensors have a residual dependency on ambient (inside the sensor housing) temperature which can be corrected for by calibrating the background samples (lowest 10%-percentile) against a reference measurement (either Picarro or network median). Delaria et al. also evaluate implications for calibration of similar networks emerging at other locations around the globe. The paper is well written, the methods are robust and rigorous, and the topic is certainly of high interest to the readers of Atmospheric Measurement Techniques. I recommend publication with minor, mostly technical modifications required.**

**Comments/questions:**
**A question I have is whether the authors made any attempt to identify the root cause for the temperature dependence. From the manuscript it becomes clear that the temperature dependencies are large, vary from sensor to sensor in terms of magnitude (and sign!), and might even be intermittent/discontinuous (L171ff). While the network analysis conceived by the authors is certainly convincing in terms of delivering an a posteriori fix to the problem, it would obviously be best if the temperature dependencies were fixed on instrument side.**

We agree with the reviewer that it would be best if temperature dependencies were fixed on the instrument side. We do not, however, try to discover or speculate as to the root cause of the problem, due to the limited availability of information on Vaisala operation from the manufacturing company. As we discuss, the problem may be more prevalent with older sensors. Our hypothesis is that it may have something to do the electrical control for separation of the optics in the Fabry-Perot interferometer, though this is purely speculation. We have added to the revised manuscript a description of the Vaisala principles of operation, in case the reader finds this useful.

L98: "The Vaisala CarboCap GMP343 instrument uses pulsed light from a filament lamp, which is reflected and refocused on an IR detector located behind a Fabry-Perot Interferometer (FPI). The FPI is electrically tuned so that its passband corresponds to either the absorption wavelength of CO2 or a reference band.

**The other question I have is whether the authors have indications for non-linearities (i.e. ΔCO₂ depending on CO₂)? Fig. 10a shows that the median of the low-cost sensors has peaks that are substantially lower than those of the Picarro. The median might not be very indicative for this particular question. But, when deploying the low-cost sensors side-by-side with the Picarro, is there a dependency of the differences on CO₂ concentration (in particular for high concentrations)?**

In L168 we discuss this possibility. We obtained corrected mixing ratios that were not statistically different if we assumed dependencies on CO2 concentrations, even for high concentrations. The median has peaks that are substantially lower than the Picarro because the Picarro makes a measurement at a particular site, while the median includes measurements from both near-urban and near-point source, and from sites more removed from city, highway, refinery emissions, etc. For example, following the temperature correction, the co-located sensor at the Richmond field station agrees with the Picarro even at high CO2 concentrations, despite that this sensor has a temperature dependence slope of 2.2. We notice certain peaks over 500 ppm where either the Picarro or the Vaisala reads higher by 1-5 ppm CO2, but this appears random and not related to the temperature.

**The manuscript is quite optimistic in claiming that the BEACO₂N achieved the required "1 ppm" accuracy (as requested by previous studies). First, all the quoted numbers (1.3  0.9ppm (L276), 1.6  0.4 ppm (L313)) are actually greater than 1 ppm. Second, the numbers might not be entirely representative of what the previous studies called the**

**"mismatch error of 1 ppm at an hourly temporal resolution"** (Turner et al., https://doi.org/https://doi.org/10.5194/acp-16-13465-2016, 2016). **To me, it seems that for the mismatch error on hourly resolution, one would actually need to combine the network error and the instrument error (and the model error which, however, is not accessible here).**

We have worked to reframe this conclusion so as to not overstate our results and to be more consistent. The below lines have been added or edited.

L253: "Minimizing the network measurement error to close to 1 ppm is desirable, as at this measurement uncertainty the error in emissions estimates from inverse modeling becomes dominated by model uncertainties (Turner et al., 2016)."

L287: "This analysis suggests that the temperature correction method provides a meaningful reduction of network measurement uncertainties toward our desired 1 ppm network error. "

L365: "We report meaningful reductions in network uncertainties following application of a temperature-dependence correction, and a resulting network instrument error of 1.6 ppm $CO_2$ or less."

**Technical comments:**

All technical corrections have been applied unless otherwise indicated. We have responded to comments below where appropriate.

**L14: " , ". Too much white space.**

**L109, equ. 1 and 4,5,6,8: The paragraph describes the temporal drift of the sensors i.e. the correction (equ.1) scales with time. I got confused by the super-script "T,drift" for the $CO_2$ offset since I wondered whether "T" (pointing to temperature) somehow anticipates/includes what follows in section 2.3. I think it might be better to just use the super-script "drift" without a "T"**

The "T" had been added in error to this section. It has been corrected.

**Equ.2: The equation lacks "x days".**

**L115: atemporal -> temporal**

Atemporal is correct. "b" is a constant offset that is constant and not dependence on time.

**L119: molybedum -> molybdenum**

**L160: "within 1  C of T(h) is calculated" Why would you need this additional constraint?**

The constraint is the basis of the temperature correction. The original time correction uses data within 1.5 weeks to calculate the 10th percentile of the data and fits this to a regression vs time to calculate the drift slope. For the temperature correction, in addition to this we find the tenth percentile of data within 1C. This is to account for that the tenth percentile at 10C may be lower than the tenth percentile at 25C, if there is indeed a positive temperature bias in the instrument.

**L169: Is the $m_T$ time-dependent? If so (e.g. in the view of "shifts dramatically" L172), using the median over the entire period might not be a good estimator.**

This is true. In this case the period is split and the median for each respective period is used. Ie. When the shift occurs the deployment period is essentially treated as two separate deployments.

**L171: "When it is observed either …." - Check sentence structure.**

**L170: Could you give a typical number for "dramatic shifts" and how often those occur?**

We have added the following sentence: "The occurrence and magnitude of this varies between instruments (0---3 times during a two year-long deployment), and is typically identified by routine checks for agreement between neighboring sensors. Shifts in the offset bias, the temperature-dependent slope, or the time-dependent drift appear as sudden or gradual offsets in mixing ratios measured by a sensor and its neighbors. Typical identified shifts in the offset bias, the temperature-dependent slope, or the time-dependent drift are on the order of 10 ppm, 0.5 ppm/C, and 1e-6 ppm/s, respectively."

**L180: Typo "calculated".**

**L193: Units missing for the two numbers at the end of the line.**

**L204: Sentence does not make sense, check sentence structure.**

**L240/241: Scale parameter? Is the scale parameter of any relevance?**

The scale parameter for a Lorentzian distribution is similar to the standard deviation in a Gaussian distribution.

**L265: For the convenience of the reader, define what a semivariance / variogramm is (e.g. write down the equation for calculating gamma_nn).**

**L275ff: I think this is a quite optimistic interpretation of the semivariance analysis: 1) the error bar on the zero-intercept is large, 2) the found 1.3  0.9 ppm is an estimate derived from a "summer months" dataset i.e. it might underestimate the error for shorter periods. Wouldn't the error estimate in section 4.2 need to be added? (See also main comment above.)**

**L293: phenomenon -> phenomena**

**Equ.9 and 10: I took me a while to understand the rationale here. Consider adding a bit more explanations.**

Sentences have been added: ""True" variations in emissions and meteorology are reflected in temporal changes in $CO_2$ concentrations due to emissions plumes and changes in wind speed and direction. Here we used temporal changes in $CO_2$ concentrations at a certain site as a proxy for "true" atmospheric variations in $CO_2$."

**L302: result is -> result in**

**L315: less that -> less than**

**Section 5: The Bay area is an area with a sea-breeze delivering pristine air on a regular basis i.e. the median 10%-percentile might work well as a background estimator. Can you say a word on whether these local conditions might be particularly favorable and whether in-land locations might have a harder time using the median background approach?**

We consider the Picarro reference instrument to serve as a representation of the regional air quality, while the dense sensor network can pick up on local signals. The median "background "approach gets at the same idea. This "background" may not actually be a background (in terms of air not directly influenced by emissions), but we believe it should still reflect the average regional non-local signal in a similar manner that the Picarro and the Bay Area median does. Essentially, although the median may not by a true "background" in Houston, it should still reflect the overall network regional average.

We have added the following sentence to L343: "The influence of a sea-breeze in the Bay Area makes the median tenth percentile $CO_2$ measured by Bay Area nodes a regional background. Although the median tenth percentile of other inland sensor networks may not represent a regional background, it can be expected to represent the overall network regional average baseline."

**Fig.10: Isn't it worrisome that panel b shows a clear time dependence?**

The time dependence does indicate that the median tenth percentile is shifting relative to the Picarro baseline. This comes largely from temporal drift in the sensors that cannot be corrected for without a reference instrument. The direction of this temporal drift would be dependent upon the temporal drift of the sensors included, which is fairly random in direction and magnitude. The error associated with this temporal drift in the median difference is included in the estimate of additional accuracy error we provide for use of the median instead of a reference instrument.

**Fig.11: The mean of the distributions seems negative. The numbers quoted in the panel titles are positive while only in the upper panel, the title says absolute value. Doublecheck whether this is all correct.**

Panel B should read absolute value of the mean. This has been corrected.

**Fig.12: Why use "fraction" difference, while throughout the manuscript "fractional" difference was used?**

This has been changed for consistency.

**Fig.Sx: In some places, the manuscript refers to Fig. Sx (indicating "supplementary" figure, I presume). There are no figures with the "S" label.**

There is a supplement where figures are given the "S" label.

**Response to Reviewer #2**

**Delaria et al present a paper on field calibrations of CO2-sensors and in particular on the correction of the temperature dependence of the sensors in a network. The authors state that temperature correction of individual sensors is necessary for achieving a good data quality. Individual sensor calibration can be done based on laboratory calibration which is, however, labor intensive and might defeat the purpose of using low-cost sensors. The authors therefore propose an in situ field calibration approach.**

**This is a well written and very relevant paper as dense sensor networks are promising for assessment, verification and tracking changes of urban CO2 emissions. This paper should be published in Atmospheric Measurement Techniques, however, I have a few issues that should be addressed:**

**- The authors mention that there is a temperature dependence in the residuals after correction of pressure and temperature effect according to the ideal gas law. This remaining temperature dependence varies for individual sensors in magnitude and sign. The authors do not provide any explanations or hypotheses on the causes of the temperature dependence, which would, however, be helpful for the reader.**

We agree with the reviewer that it would be very helpful to know what the source of this temperature dependence is. We do not, however, try to discover or speculate as to the root cause of the problem, due to the limited availability of information on Vaisala operation from the manufacturing company. As we discuss, the problem may be more prevalent with older sensors. Our hypothesis is that it may have something to do the electrical control for separation of the optics in the Fabry-Perot interferometer, though this is purely speculation. We have added to the revised manuscript a description of the Vaisala principles of operation, in case the reader finds this useful. We leave it to the reader to form their own hypotheses.

**- Figure 2 shows an example of the temperature dependence of a CO2 sensor. There is a very strong linear temperature dependence. Figure 3 shows another example of a strong and this time non-linear temperature dependence of opposite sign. In both examples, there is of course some variability in the derived temperature**

**dependence, e.g. caused by comparison of CO2 as measured at two distant locations (deployment and reference site). However, this leads to some uncertainty in the parameter estimation for the temperature correction. The authors therefore should provide uncertainty calculations for the temperature correction. I expect that at least for some sensors, the uncertainty in the parameter estimation for the temperature correction is around or larger than the ambitious data quality goal (1ppm). Actually, consideration of uncertainty in the data correction method is completely missing (also for drift correction) and should be included. Specifically, uncertainty considerations should be made for the calculations according to equations 6 and 8. It would then be interesting to see how the uncertainty of individual corrected CO2 sensors compares to the network error as estimated later in the paper. The strong temperature dependence of sensors as shown in Figs. 2-6 give the impression that the calculated network error is too optimistic.**

**- The authors claim, albeit implicitly, that field calibrations lead to similar performance than sensors calibrated in the laboratory. It would have been interesting to demonstrate this by deploying laboratory calibrated CO2 sensors and comparing the data postprocessed using the the two different calibration approaches. I know this is too much for now, but would be insightful in the future.**

We agree that this would have been a great and very insightful experiment. Limitations with accessing sensors in the field due to the Covid-19 pandemic, and other time limitations, resulted in us unfortunately being unable to test sensor temperature dependence in the laboratory.

The following line edits have all been made unless otherwise indicated. We have commented where appropriate.

**P5, L139/140: Sentence is linguistically not correct, please correct.**

**P6, L. 171: "… the either the …" correct.**

**P6, equations 4 and 5. med_mT is not defined. mT has been defined, but it has not been stated that med_mT is the median of the slope.**

**P7, L. 198: "… and without and adjustment …" delete second "and".**

**P8, L. 223. The authors mention that results were identical when using a multiplicative correction term. It is difficult to understand what this exactly means. How has a multiplicative correction term be determined, how does the equations look like? The authors should provide more details (e.g. in the supplementary Information).**

This had been explained in more detail on L166.

**P9, L. 265. The authors evaluated the network error based on a semivariogram. A sentence what a semi-variogram is and the underlying idea would be helpful.**

We have added the equation for semivariance following suggestion by reviewer #1. ",…of gamma_nn vs distance,…" was also added to this sentence.

**P11, equations 11 and 12. Notation can be improved, it is unclear which index is used for summation.**

**P11, L. 324. It is not defind what "STP" stands for. Should be mentioned.**

Variable had been defined in L102. A further definition of STP has been added to this line.

**Fig 6a. The Picarro signal is not visible in the plot for 2018.**

This has been fixed.

**Figure S4. Missing data are filled with straight lines (orange and red), should be corrected.**

This has been fixed.